# Mechanism of outer membrane destabilization by global reduction of protein content

Irina V. Mikheyeva [1,5], Jiawei Sun[2,5], Kerwyn Casey Huang [2,3,4] & Thomas J. Silhavy [1]

The outer membrane (OM) of Gram-negative bacteria such as *Escherichia coli* is an asymmetric bilayer with the glycolipid lipopolysaccharide (LPS) in the outer leaflet and glycerophospholipids in the inner. Nearly all integral OM proteins (OMPs) have a characteristic β-barrel fold and are assembled in the OM by the BAM complex, which contains one essential β-barrel protein (BamA), one essential lipoprotein (BamD), and three non-essential lipoproteins (BamBCE). A gain-of-function mutation in *bamA* enables survival in the absence of BamD, showing that the essential function of this protein is regulatory. Here, we demonstrate that the global reduction in OMPs caused by BamD loss weakens the OM, altering cell shape and causing OM rupture in spent medium. To fill the void created by OMP loss, phospholipids (PLs) flip into the outer leaflet. Under these conditions, mechanisms that remove PLs from the outer leaflet create tension between the OM leaflets, which contributes to membrane rupture. Rupture is prevented by suppressor mutations that release the tension by halting PL removal from the outer leaflet. However, these suppressors do not restore OM stiffness or normal cell shape, revealing a possible connection between OM stiffness and cell shape.

Gram-negative bacteria such as *Escherichia coli* are protected from the environment by their cell envelope, which is made unique by the outer membrane (OM). The OM, an asymmetric bilayer with lipopolysaccharide (LPS) in the outer leaflet and glycerophospholipids (PLs) in the inner leaflet, functions as a selective permeability barrier, allowing passage of small hydrophilic compounds while excluding large and hydrophobic molecules[1]. An ordered network of outer membrane proteins (OMPs) with patches of LPS stabilized by electrostatic LPS-LPS and LPS-OMP interactions contribute to the mechanical strength of the cell[2–4]. Perturbation to this essential barrier can result in cell death.

The integrity of the OM relies on proper localization of its multiple components. LPS is transported by the Lpt machinery, which is composed of seven proteins that form a trans-envelope bridge[5]. Asymmetry is then established by multiple mechanisms. First, the Mla

system shuttles mislocalized PLs back to the inner membrane (IM)[6]. Second, the outer membrane phospholipase PldA breaks down mislocalized PLs and acts as a sensor of OM asymmetry[7,8]. While LPS transport and maintenance of OM asymmetry are well understood, PL transport to the OM remains poorly characterized. However, recent work indicates that PLs are transported from the IM to the inner leaflet of the OM by protein bridges using redundant AsmA-like proteins such as YhdP, TamB, and YdbH[9–12].

OMPs are typically β-barrel proteins that span the OM and are assembled by the β-barrel assembly machine (BAM) complex. The *E. coli* BAM complex is heteropentomeric, containing two essential proteins, BamA and BamD, and three non-essential proteins, BamB, BamC, and BamE. BamA, itself a β-barrel protein, catalyzes the folding of OMPs, while the lipoprotein BamD plays a regulatory and substrate

[1]Department of Molecular Biology, Princeton University, Princeton, NJ 08540, USA. [2]Department of Bioengineering, Stanford University, Stanford, CA 94305, USA. [3]Department of Microbiology and Immunology, Stanford University School of Medicine, Stanford, CA 94305, USA. [4]Chan Zuckerberg Biohub, San Francisco, CA 94158, USA. [5]These authors contributed equally: Irina V. Mikheyeva, Jiawei Sun. ✉e-mail: kchuang@stanford.edu; tsilhavy@princeton.edu

quality control role in OMP assembly[13,14]. Previous work identified an allele of *bamA*, *bamA[E470K]*, that is viable in the absence of *bamD*[14]. Deletion of *bamD* in the *bamA[E470K]* background results in increased OM permeability[14], but the structural changes underlying this phenotype are not well understood.

In this work, we investigate the consequences of *bamD* deletion on the structural integrity of the OM and on cell shape. *bamD* deletion resulted in a highly inefficient BAM complex, leading to decreased OMP levels and mislocalization of PLs to the outer leaflet of the OM[14]. We show that these structural changes alter growth rate and cell shape and cause a striking cell death phenotype, precipitated by rupture of the OM in spent medium or under oscillatory hyperosmotic shocks. Suppressor analysis suggested that mechanisms that remove PLs from the OM outer leaflet under protein-depleted conditions create interleaflet tension that destabilizes the OM. Relieving this tension through suppressor mutations rescued cell death, but altered cell shape and OM mechanical stiffness remained compromised, emphasizing the central role of OM proteins and OM stiffness in cellular morphogenesis and integrity.

## Results

### Deletion of *bamD* decreases fitness in stationary phase and causes OM permeability defects

A previous study showed that *bamA[E470K]* cells can tolerate deletion of *bamD*[14]. Given the otherwise essential role of *bamD*, we sought to further characterize the effects of *bamD* deletion on cellular physiology. As observed previously, overnight cultures of a *bamA[E470K] ΔbamD* strain (hereafter, *ΔbamD*) exhibited lower optical density (OD) than the parent *bamA[E470K]* strain[14], and *ΔbamD* cultures exhibited lower maximum growth rate and pronounced lysis during stationary phase compared with the parent (Fig. 1a). To further interrogate the stationary phase lysis of *ΔbamD* cultures, we resuspended log-phase cells in LB supernatant collected by filtering the cells from an overnight wild-type culture ("spent medium"), which lacks the nutrients needed to support growth and rapidly transitions cells into stationary phase. This spent medium assay was used previously to probe membrane integrity in cells with the *mlaA** allele[15], which disrupts OM lipid homeostasis. *ΔbamD* and MC4100 *mlaA** cells exhibited nearly identical lysis dynamics upon transitioning to spent medium, while the OD of the parent remained high (Fig. 1b). Since spent LB lacks the $Mg^{2+}$ necessary to stabilize LPS-LPS interactions, we reasoned that the weakened OM of the *ΔbamD* mutant was not able to withstand the additional stress. Indeed, when we supplemented spent LB with $Mg^{2+}$, lysis was abolished (Fig. S1).

*ΔbamD* colonies were very mucoid, suggesting that the Rcs stress response is activated due to perturbations to the OM. To probe the integrity of the OM, we plated parent and *ΔbamD* cells on LB and LB supplemented with bacitracin, a large hydrophilic peptide antibiotic that is typically excluded from *E. coli* cells[16], or SDS-EDTA. *ΔbamD* cells were unable to grow in the presence of bacitracin or SDS-EDTA (Fig. 1c). To further test OM permeability, we quantified uptake of fluorescent 1-N-phenyl-naphthylamine (NPN)[17]. After 20 min, wild-type MC4100 and *bamA[E470K]* cells displayed much lower uptake than *ΔbamD* cells (Fig. 1d), indicating that *bamD* deletion generally increased OM permeability. Together, these results indicate that *bamD* deletion dramatically alters the OM.

### *bamD* deletion reduces growth rate and disrupts cell shape

We noted that deletion of *bamD* decreased the maximal growth rate in exponential phase (Fig. 1a), and sought to characterize the effect of *bamD* deletion on cell shape and single-cell growth dynamics. Using time-lapse imaging, we tracked single cells from log-phase cultures on an LB agarose pad and quantified single-cell growth rates (Fig. 2a, b). *ΔbamD* cells exhibited disrupted shapes, usually featuring more non-uniform cell width along the cell body and greater elongation compared with the parent (Fig. 2a, Movie S1, S2), indicating disrupted

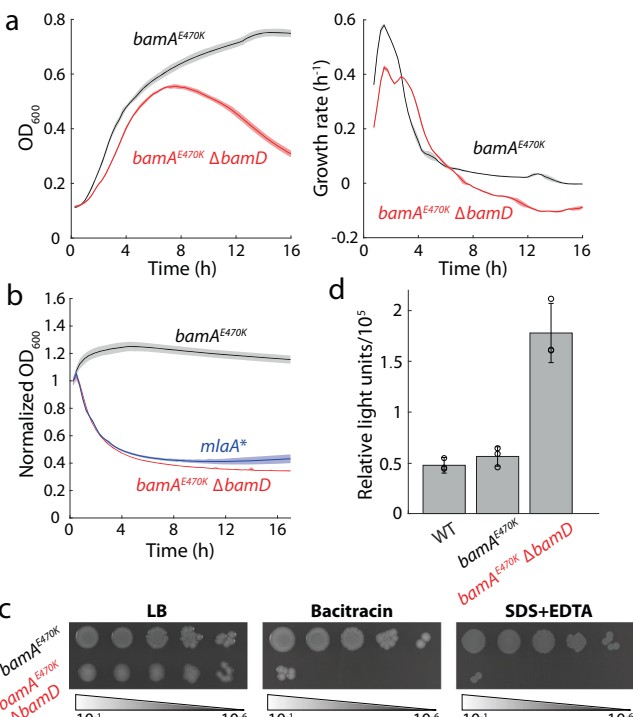

**Fig. 1 | Deletion of *bamD* leads to cell death during stationary phase and a highly permeable membrane. a** *bamD* deletion leads to slower growth in log phase and a decrease in $OD_{600}$ in stationary phase compared with the parent strain *bamA[E470K]*. Left: $OD_{600}$ measurements starting from a diluted overnight culture. $N = 3$ biological replicates. Curves are mean values, and shaded regions represent 1 standard deviation. Right: corresponding growth rates as quantified by $d\ln(OD)/dt$. **b** *ΔbamD* and *mlaA** cells exhibited similar decreases in OD when transferred from log phase to the spent medium of wild-type *E. coli* MC4100, while the parent strain maintained a stable OD. All OD measurements were normalized to the value upon transfer to spent medium at $t = 0$. $N = 3$ biological replicates. Curves are mean values, and shading represents 1 standard deviation. **c** *ΔbamD* cells exhibited inhibited growth on plates with bacitracin or SDS-EDTA compared with LB plates, whereas growth of the parent strain was not affected by either treatment. Data are representative of three biological replicates. **d** *ΔbamD* cells exhibited increased OM permeability compared with the parent strain as measured by the intake of fluorescent 1-N-phenyl-naphthylamine (NPN), normalized by $OD_{600}$. Data are presented as mean values ± 1 standard deviation. $N = 3$ biological replicates.

shape regulation. The average instantaneous growth rate of *ΔbamD* cells was significantly lower than that of wild-type cells even when controlling for similar cell shape (Fig. 2b), consistent with population growth measurements (Fig. 1a). Moreover, instantaneous growth rate decreased as the cell-width coefficient of variation increased (Fig. 2b), suggesting that deletion of *bamD* causes a metabolic burden linked to envelope disruption. Some cells eventually lysed (Fig. 2a, arrows), signifying weakening of the cell envelope. Additionally, we quantified the morphology of two other BAM complex mutants: *bamA101*, a transposon insertion in the *bamA* promoter that decreases BamA levels by >90%, and deletion of *bamB*, which encodes a nonessential BAM complex lipoprotein. We selected *ΔbamB* because this mutant has greater permeability defects than *ΔbamC* or *ΔbamE* mutants[18–20]. The *bamA101* and *ΔbamB* mutants showed disrupted regulation of cell width compared to *bamA[E470K]* (Fig. S2a). Thus, disrupting the BAM complex has global effects on cell morphology and growth.

### The OM ruptures prior to the IM in *ΔbamD* cells when transitioned to spent medium

Based on the similarity of lysis dynamics upon starvation in *mlaA** cells and *ΔbamD* cells (Fig. 1b), we hypothesized based on our previous

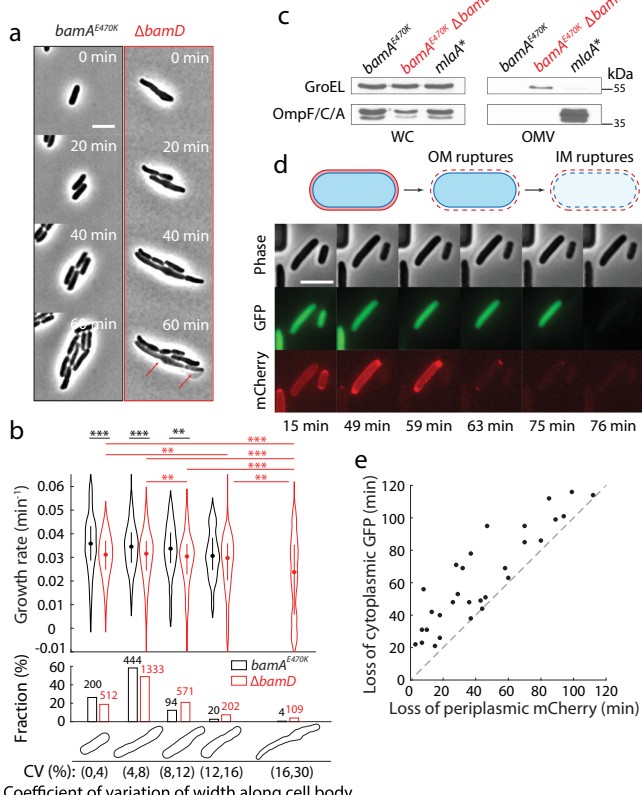

**Fig. 2 | Δ*bamD* cells exhibit defects in growth rate and cell shape, and the outer membrane ruptures before the inner membrane in spent medium. a** As compared with the parent strain, Δ*bamD* cells exhibit more variation in cell width along the cell body as well as lysis (red arrows). Shown are time-lapse images of log-phase cells on an agarose pad with LB. Scale bar: 5 μm. **b** Δ*bamD* cells had lower instantaneous growth rates ($d\ln A/dt$, where $A$ is cell area; top) compared with the parent strain. Shown are cells grouped based on the degree of cell shape defect (bottom), as quantified by the coefficient of variation of width along the cell body (Methods). Error bars represent 25th and 75th percentiles. Example cell contours for each grouping are shown under the corresponding bar plots. The number of cells in each group is shown on the barplot. Two-sample Kolmogorov-Smirnov test; **: $p < 0.01$, ***: $p < 0.001$. Exact $p$-values are included in Supplementary Table 2. **c** Immunoblots of OM porins in the whole cell (WC) or OM vesicles (OMVs) of *bamA*^E470K, Δ*bamD*, and *mlaA** cultures, compared to the level of the intracellular protein GroEL used as a control. Deletion of *bamD* does not induce OMV production, by contrast to the *mlaA** allele. Data are representative of three biological replicates. **d** In a Δ*bamD* cell shifted from log-phase growth in LB to spent LB, the OM ruptures before the IM. Shown are time-lapse images of Δ*bamD* cells expressing cytoplasmic GFP and periplasmic mCherry incubated in a microfluidic flow cell (Methods). The longer cell on the left maintained cytoplasmic GFP for >15 min after periplasmic mCherry was lost. Scale bar: 5 μm. **e** For all cells measured as in (**d**), periplasmic mCherry was lost earlier or at the same time as cytoplasmic GFP in Δ*bamD* cells shifted from log phase growth in LB to spent LB. $n = 30$ cells.

findings[15] that *bamD* deletion leads to loss of outer membrane material via vesiculation. To test this hypothesis, we collected outer membrane vesicles (OMVs) from exponentially growing cells (Methods). While we were able to easily harvest OMVs from *mlaA** cells, preparations from parent or Δ*bamD* cells displayed no detectable OM proteins (Fig. 2c). Instead, in Δ*bamD* cells we detected the cytoplasmic protein GroEL, suggesting leakiness due to cell rupture during the centrifugation used for OMV harvest.

To identify the mechanism by which Δ*bamD* cells lyse in spent medium, we used a microfluidic flow cell to rapidly transition log-phase cells to spent medium (Methods). Motivated by our previous study in which we showed that the IM ruptures before the OM in *mlaA**

cells[11], we utilized a Δ*bamD* strain that constitutively expresses cytoplasmic GFP and periplasmic mCherry (Table S1). Time-lapse imaging revealed that periplasmic mCherry fluorescence was always lost prior to (Fig. 2d) or coincident with the loss of cytoplasmic GFP (Fig. 2e), indicating that the OM ruptured before the IM. Together, these data indicate that a severely destabilized OM is the direct cause of Δ*bamD* cell lysis in stationary phase.

## OM composition is altered by *bamD* deletion

To further interrogate the lysis phenotype, we collected cell lysate from cells grown at 37 °C in log phase (Methods) and measured the abundance of OM components such as the major OM porins (Fig. 3a, lane 1 and 2; Fig. 3b, lane 1 and 2) and LPS (Fig. 3b, lane 1 and 2). Deletion of *bamD* dramatically decreased levels of OM porins, including the essential BamA and LptD. By contrast, LPS levels were relatively unaffected by *bamD* deletion even though LptD levels were decreased (Fig. 3b, lane 1 and 2). Taken together, these results indicate that the OM of Δ*bamD* cells is highly lacking in proteins, and that lysis is not due to membrane loss but instead due to altered OM composition.

## Mutants that increase PL levels in the OM outer leaflet suppress lysis due to *bamD* deletion

Efforts to isolate spontaneous suppressors of Δ*bamD* lysis were unsuccessful, hence we attempted to identify rescue conditions for growth and decreased lysis through strategic genetic manipulation of the OM. In previous work, we identified the OM outer leaflet phospholipase PldA as a potent suppressor of *mlaA** lysis[15]. Deletion of *pldA* from Δ*bamD* cells slowed down lysis kinetics in both stationary phase and the spent medium assay, and moderately improved the initial growth dynamics (Fig. 3c). We reasoned that since Δ*bamD* cells are lacking in many proteins that typically make up a large fraction of the OM, deletion of *pldA* allowed the cells to retain more PLs to fill the voids. Thus, by analogy with *Acinetobacter baumannii* mutants that lack LPS, we reasoned that other mutations that allow PLs to accumulate in the OM outer leaflet might also act as suppressors of Δ*bamD* lysis[21,22]. As shown in previous studies, inhibiting the Mla transport system and deleting *pldA* increases PL levels in the OM outer leaflet by ~25-fold and drastically changes OM physiology[2,6]. We found that deletion of *mlaA* from Δ*bamD* cells was not sufficient to rescue lysis (Fig. 3c). Nonetheless, lysis was virtually abolished in Δ*bamD* Δ*pldA* Δ*mlaA* cells in both stationary phase and spent medium (Fig. 3c). Single-cell lysis dynamics of the suppressors Δ*bamD* Δ*pldA*, Δ*bamD* Δ*mlaA*, and Δ*bamD* Δ*pldA* Δ*mlaA* upon exposure to spent medium (Fig. 3d) closely mimicked population-level measurements (Fig. 3c).

The *lpxC101* allele decreases lipid A biosynthesis[23,24] and frees up precursors for PL synthesis, leading to increased levels of PLs in the outer leaflet of the OM[25]. Introduction of *lpxC101* to Δ*bamD* cells improved growth dynamics and ameliorated lysis similar to the deletion of *mlaA* (Fig. S3a). Moreover, the combination of *lpxC101* with Δ*bamD* Δ*pldA* restored growth dynamics and final yield to approximately that of the *bamA*^E470K parent and lysis in stationary phase and the spent medium assay was completely abolished (Fig. S3a, b). Δ*bamD* *lpxC101* Δ*mlaA* cells still showed lysis in spend medium, suggesting that PldA activity is detrimental in the Δ*bamD* background (Fig. S3c).

To investigate the basis for rescue, we first tested whether the Bam complex was functioning more efficiently in the suppressor backgrounds. The Δ*bamD* Δ*mlaA* Δ*pldA* and Δ*bamD* *lpxC101* Δ*pldA* suppressors of lysis did not restore OMP levels to that of wild type, nor did any of the other single mutants (Fig. 3b). Moreover, levels of oxidized or reduced LptD showed a similar pattern in all mutants, and LPS levels were only reduced in strains with the *lpxC101* allele (Fig. 3b). As noted above, the Δ*bamD* colonies are mucoid, indicating production of the colanic acid capsule. Since *E. coli* K-12 strains lack an O-antigen,

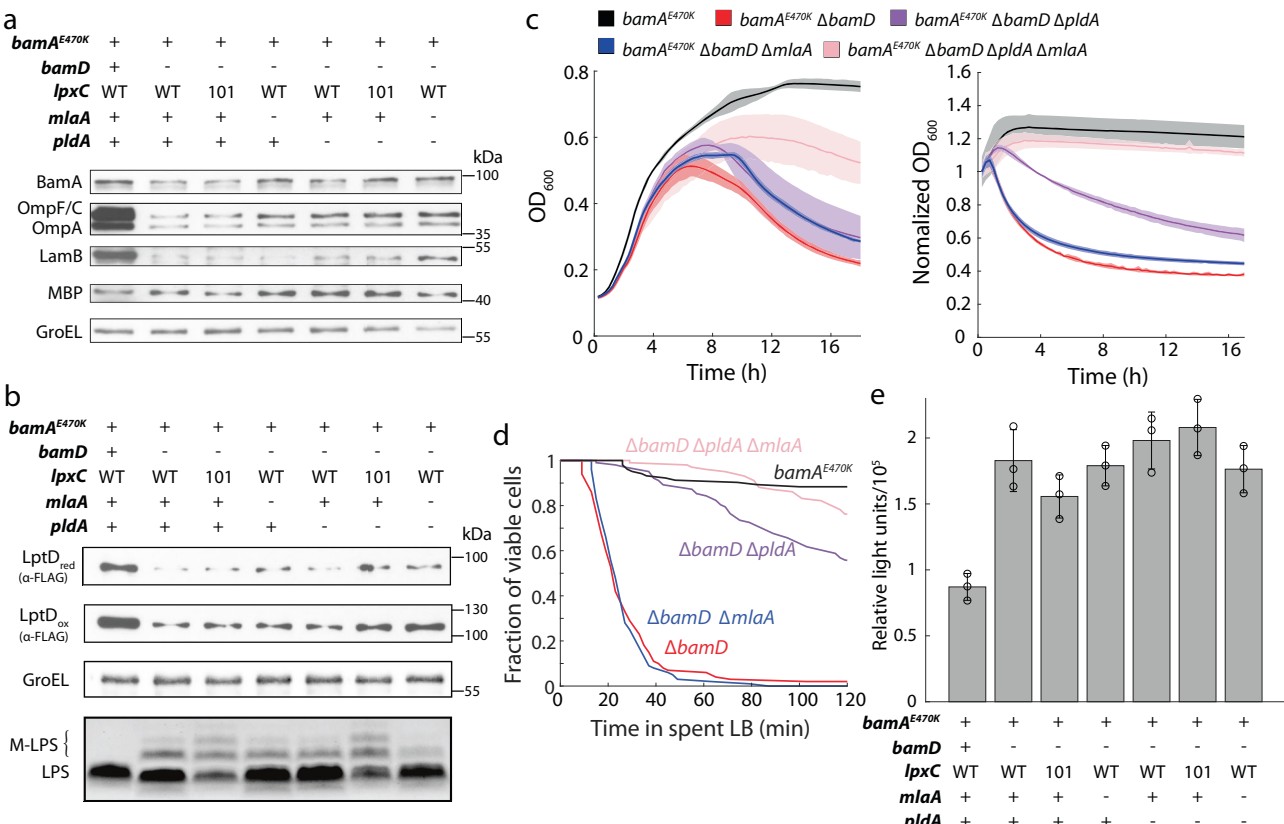

**Fig. 3 | OM lysis is not due to loss of material but instead due to alteration in membrane composition. a** *bamD* deletion reduces levels of OM proteins. Shown are immunoblots of major OM proteins in cell lysates of *bamA*[E470K], Δ*bamD*, and suppressor mutants of Δ*bamD* from log-phase cultures. Data are representative of three biological replicates. **b** Although deletion of *bamD* reduces levels of the LPS transport protein LptD, LPS levels are unaffected. Shown are immunoblots of LptD, GroEL (as a control), and LPS in cell lysates of *bamA*[E470K], Δ*bamD*, and suppressor mutants of Δ*bamD* from log-phase cultures. Data are representative of three biological replicates. **c** The decrease in OD in stationary phase (left) or upon transition from log-phase growth to spent LB (right, transition occurs at *t* = 0) in Δ*bamD* cells is partially rescued by deletion of *pldA* and fully rescued by deletion of *pldA* and

*mlaA*. *N* = 3 biological replicates. Curves are mean values, and shading represents 1 standard deviation. **d** Cell lysis in spent LB is partially rescued by deletion of *pldA* and almost fully rescued by deletion of *pldA* and *mlaA*, in agreement with the population-level measurements in (**c**). Shown are quantifications of the fraction of viable cells as determined from time-lapse images (Methods). *n* > 100 cells for each strain. **e** Suppressors do not rescue the permeability defect of Δ*bamD* cells. Shown is quantification of the intake of fluorescent 1-N-phenyl-naphthylamine (NPN) in *bamA*[E470K], Δ*bamD*, and suppressor mutants. Data are presented as mean values ± 1 standard deviation. *N* = 3 biological replicates. Relative light units = fluorescence/OD_{600}.

the O-antigen ligase attaches colanic acid to LPS, producing a larger, modified form of LPS. This modification did not affect the lysis phenotype, as disruption of the upstream Rcs pathway did not reduce the degree of lysis (Fig. S4).

The Δ*bamD* Δ*mlaA* Δ*pldA* and Δ*bamD* *lpxC101* Δ*pldA* suppressor mutants had a similar increase in permeability in our NPN assay relative to the parent (Fig. 3e). Collectively, these data suggest that prevention of lysis depends on PL levels in the OM outer leaflet, and not restoration of protein networks or LPS modification.

### Populating the OM outer leaflet with PLs does not restore cell shape or viability under large osmotic shocks

One of the essential functions of the bacterial OM is to provide mechanical integrity for withstanding forces from inside and outside of the cell[3,26]. The stiffness of the OM is largely determined by molecular interactions among outer membrane components including proteins and LPS molecules and between the OM and the cell wall[2–4]. Based on the altered OM composition of the Δ*bamD* strain and its suppressors (Fig. 3a, b), we hypothesized that a reduction in OM protein levels or increased PLs would affect the stiffness of the OM. To quantify sensitivity to mechanical perturbations, we performed oscillatory hyperosmotic shock assays in a microfluidic flow cell (Methods)[3,27]. Δ*bamD* cells exhibited a drastic reduction in viability

compared to the parent strain (Fig. 4a, Movie S3). While Δ*bamD* Δ*mlaA* Δ*pldA* cells did not exhibit lysis in spent medium (Fig. 3c, d), they were still more sensitive to large osmotic shocks than the parent strain (Fig. 4a).

Previously, we quantified the stiffness of the OM in various Gram-negative bacteria using a plasmolysis/lysis assay in which cells were exposed first to a large hyperosmotic shock to remove turgor pressure and then a detergent or EDTA to lyse the cells (Methods)[3]. Consistent with our previous findings, cell length of the parent strain decreased by ~12% upon plasmolysis and then by a further ~11% upon lysis (Fig. 4b), indicating that compression of the stiff outer membrane was holding the cell wall in an extended state during plasmolysis. By contrast, Δ*bamD* and Δ*bamD* Δ*pldA* Δ*mlaA* cells shrank by ~30% upon plasmolysis and exhibited very little further contraction upon lysis (Fig. 4b), suggesting that the OM in these strains is much weaker than the parent strain. Interestingly, the stiffness of the OM in the *bamA101* and Δ*bamB* mutants was similar to that of *bamA*[E470K] cells, as indicated by the comparable cell length contraction upon plasmolysis and subsequent cell lysis (Fig. S2b).

Similar to a spring, contraction upon plasmolysis is determined by a combination of the stiffness and rest length (amount of material) of the OM. We previously showed in wild-type cells that the OM rest length is similar to the length of the turgid cell[3], indicating that there is

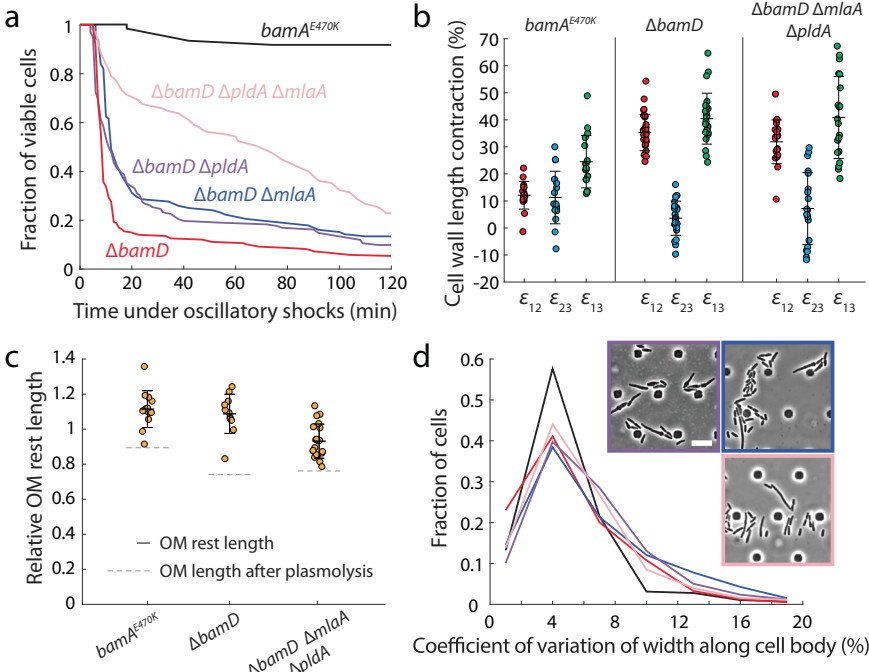

**Fig. 4 | Replacement of proteins with PLs does not restore cell shape or OM stiffness. a** Lysis due to oscillatory hyperosmotic shocks (cycles of 1 min in LB and 1 min in LB + 400 mM sorbitol; Methods) is partially rescued by deletion of both *pldA* and *mlaA*. Shown are measurements of the fraction of viable cells from time-lapse images in a microfluidic flow cell. $n = 60$ for *bamA*[E470K]; $n > 100$ cells for all other strains. **b** Δ*bamD* and Δ*bamD* Δ*pldA* Δ*mlaA* cells have lower OM stiffness than the parent strain. Shown are measurements of cell wall length contraction from the turgid state to plasmolyzed state after addition of 1 M sorbitol ($\varepsilon_{12}$), from the plasmolyzed state to lysed state after addition of detergent ($\varepsilon_{23}$), and from the turgid state to the lysed state ($\varepsilon_{13}$) (Methods). $\varepsilon_{12}$ and $\varepsilon_{23}$ are higher and lower, respectively, in Δ*bamD* and Δ*bamD* Δ*pldA* Δ*mlaA* compared with the parent strain, suggesting that the OM in the two mutants is not as able to resist contraction upon turgor release. Data are presented as mean values ±1 standard deviation. Sample sizes: $n = 17$ for *bamA*[E470K], $n = 24$ for Δ*bamD* (plus 6 cells for $\varepsilon_{12}$ only), $n = 20$ for Δ*bamD* Δ*pldA* Δ*mlaA*. **c** The OM rest length is similar to that of the turgid cell in *bamA*[E470K], Δ*bamD*, and Δ*bamD* Δ*pldA* Δ*mlaA* cells, indicating that the OM of all strains have a similar amount of material. Shown is the ratio of the OM size of a spheroplast compared to the turgid OM area (Methods). In all three strains, the OM rest length is substantially larger than a cell after plasmolysis (ratio of OM length after plasmolysis to the turgid length is shown as grey dashed lines). $N = 1$ biological replicate. Data are presented as mean values ±1 standard deviation. Sample sizes: $n = 13$ for *bamA*[E470K], $n = 11$ for Δ*bamD*, $n = 20$ for Δ*bamD* Δ*pldA* Δ*mlaA*. **d** Cell shape defects due to *bamD* deletion are not rescued in the suppressor strains Δ*bamD* Δ*pldA*, Δ*bamD* Δ*mlaA*, and Δ*bamD* Δ*pldA* Δ*mlaA*. All three strains exhibited higher coefficient of variation in cell width along the cell body compared to the parent. Lines are colored by strain as in (**a**). $n > 200$ cells for each strain. Insets: cells were imaged during log-phase growth in a microfluidic flow cell. Scale bar: 10 μm.

little stress in the OM during steady-state growth. Mimicking these findings, we found that OM rest length in *bamA*[E470K], Δ*bamD*, and Δ*bamD* Δ*pldA* Δ*mlaA* cells was similar to the length of the turgid cell and significantly larger than the plasmolyzed length, which is ~75% of the turgid cell length (Fig. 4c, Methods). In other words, even though the OM of Δ*bamD* cells lacks OMPs, the amount of material in the OM is not substantially lower than in wild-type cells.

By assuming that the OM rest length is equivalent to the turgid length, we estimated the mean OM stiffness (Methods) to be ~90% and ~75% lower in Δ*bamD* and Δ*bamD* Δ*pldA* Δ*mlaA* cells, respectively, compared with *bamA*[E470K] cells. Moreover, exponentially growing Δ*bamD* Δ*pldA*, Δ*bamD* Δ*mlaA*, and Δ*bamD* Δ*mlaA* Δ*pldA* cells all exhibited disrupted cell shapes (Fig. 4d), similar to Δ*bamD* cells (Fig. 2a). Taken together, we infer that reduction in the levels of OM proteins mechanically destabilized the OM, and increasing the levels of PLs by deleting *pldA* and *mlaA* does not restore mechanical integrity.

## Discussion

Our results demonstrate that a global decrease in OMP levels resulting from lack of BamD in the BAM complex compromises cell viability and shape. The destabilized OM ruptures during stationary phase due to increased tension and has significantly decreased stiffness with which to withstand mechanical stress.

As observed previously with *A. baumannii*[21,22], to populate voids in the OM caused by the lack of OM components in *E. coli* Δ*bamD* cells, PLs flip from the inner leaflet of the OM to the outer leaflet. In the case of *A. baumannii*, the OM component lacking was LPS, while *E. coli* Δ*bamD* it was OMPs. In both organisms, PldA and Mla pathway activityes are detrimental, and both activities must be removed to relieve the defects. It is likely that the OMP defect is more severe than the LPS defect. *A. baumannii* strains lacking LPS exhibit growth defects[22], while in *E. coli* Δ*bamD*, the OMP defects both slow growth and cause cell lysis in spent media.

In vitro studies with purified PldA suggest that the enzyme binds diacyl compounds three to five times better than monoacyl compounds, demonstrating a preference for PLs[28]. Thus, in an envelope that has an abundance of PLs, the majority of PldA activity may be generating lyso-phospholipids rather than fully breaking down the PLs. In turn, lyso-phospholipids have been implicated in changes to local membrane properties and in modification of embedded protein function[29,30]. We suggest that the decreased production of these lyso-phospholipids explains why removing PldA has a greater effect on cellular viability of the Δ*bamD* mutant than does removal of the Mla transport system.

Gram-negative bacteria possess several mechanisms for maintaining lipid asymmetry in the OM and as such, there is likely no mechanism for flipping PLs from the inner leaflet to the outer leaflet; such an activity would be counterproductive. In strains where PLs are needed in the outer leaflet to fill the void created by the absence of one of the major OM components, uncatalyzed PL flipping happens slowly and the resulting imbalance in material content creates tension between the membrane leaflets (Fig. 5). To make matters worse, PLs

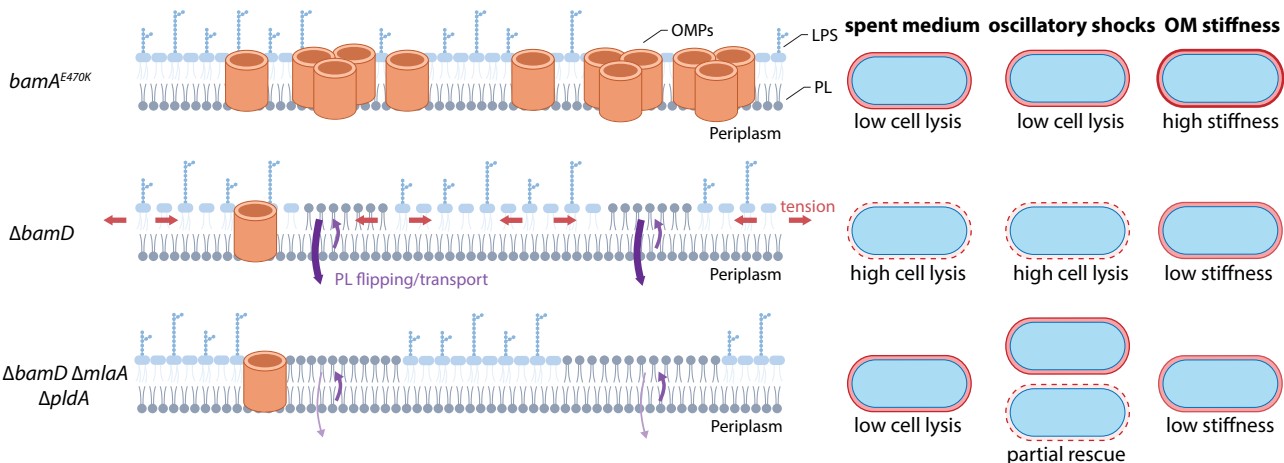

**Fig. 5 | Model of OM destabilization by the severe reduction in protein content due to *bamD* deletion.** When Δ*bamD* cells are transferred from log-phase growth to spent LB or exposed to oscillatory osmotic shocks, they exhibit much higher rates of cell lysis than the parent *bamA*^*E470K* strain, due to a reduction in OMP levels and consequent decrease in OM stiffness. Deletion of *mlaA* and *pldA* increases the amount of PLs in the outer leaflet of the OM, which relieves interleaflet tension and prevents lysis in spent medium. However, the Δ*bamD* Δ*mlaA* Δ*pldA* mutant exhibits only partial rescue of lysis under oscillatory osmotic shock and OM stiffness remains low.

are further limited in the outer leaflet via rapid degradation by PldA and the activity of the Mla transport system (Fig. 5). We hypothesize that by populating the outer leaflet with PLs, interleaflet membrane tension is decreased, growth defects are ameliorated, and OMP-depleted *E. coli* cells can survive in nutrient- and cation-depleted spent LB (Fig. 5). Interestingly, while the growth defects and lysis can be suppressed in the Δ*pldA* Δ*mlaA* double mutant, the suppressor mutations do not restore mechanical integrity, the OM permeability barrier, or cell shape (Fig. 5).

Together, our data combined with the previous studies of LPS- or lipooligosaccharide (LOS)-deficient *A. baumannii* suggest that the intact, ordered OMP network stabilized by LPS-LPS and LPS-protein interactions[2–4] are required to form a stiff OM. Our previous study implicated both LPS and OMPs as partial contributors to OM stiffness[3]. The reduction of OMP levels in Δ*bamD* cells destroys the ordered OMP network and LPS presumably forms disconnected islands, leading to almost complete loss of OM stiffness. Proposed mechanisms of cell shape determination in bacteria tend to focus on the peptidoglycan cell wall due to its solid-like structure and ability to maintain its shape in the absence of other components[31]. However, our suppressor mutants imply the existence of a connection between OM stiffness and cell shape regulation, with a stiff OM mechanically restraining cell width fluctuations during growth and preventing the development of irregular cell shapes. PBP1A glycosyltransferase activity is lethal for LOS-deficient *A. baumannii* strains, suggesting that it is PBP1A activity that is affected by OM stiffness, leading to the observed irregular cell shapes in Δ*bamD* cells[32]. The ability to perturb specific components of the complex OM composition, in this case OMPs through *bamD* deletion, thus provides insight into a distinct physiological state in which new mechanisms of cell envelope regulation can be revealed.

## Methods

### Bacterial strains and growth conditions
Strains, plasmids, and oligonucleotides are listed in Table S1. Strains were grown in lysogeny broth (LB) supplemented with 25 mg/L kanamycin, 25 mg/L ampicillin, or 25 mg/L tetracycline when appropriate. Strains were maintained at 30 °C to decrease the frequency of lysis, while experimental data were acquired at 37 °C. *bamA*^*E470K* was generated at the native *bamA* locus through recombineering using *Collinsella stercoris* phage RecT (CspRecT) as described previously[33]. All other chromosomal mutations were constructed using generalized transduction[34]. Deletions originated from the Keio collection[35].

Kanamycin resistance cassettes were excised using the FLP recombinase as described previously[36].

### Growth curves
Overnight cultures were diluted 1:100 in fresh LB and grown at 30 °C for 2 h. Optical density was measured, and cells were diluted to $OD_{600} = 0.05$ in 500 mL of LB in a 24-well plate (CytoOne, USA Scientific, Cat. #CC7682-7524). The plate was covered with a Breathe-Easy gas-permeable membrane (Sigma-Aldrich, Cat. #Z380059) and grown overnight in a Synergy H1 microplate reader (BioTek) at 37 °C with continuous orbital shaking at 559 cpm frequency. $OD_{600}$ was measured every 15 min for 18 h.

### Lysis curves
To make spent media, the wild-type MC4100 strain was grown in LB medium overnight, followed by centrifugation and filtration through a 0.2-μm filter to create cell-free spent medium. Early exponential phase cells ($OD_{600} = 0.6$–$0.8$) were concentrated to $OD_{600} = 1$ via centrifugation. Supernatant was discarded and the cell pellet was resuspended in 1 mL of spent medium, transferred to a 24-well plate (CytoOne, USA Scientific, Cat. #CC7682-7524), and covered with a Breathe-Easy gas-permeable membrane (Sigma-Aldrich, Cat. #Z380059). Lysis was monitored by decrease in optical density in a Synergy H1 microplate reader (BioTek) at 37 °C with continuous orbital shaking at 559 cpm frequency. $OD_{600}$ was measured every 15 min for 18 h.

### Efficiency of plating assay
Exponential phase cultures were normalized to $OD_{600} = 1$ and serially diluted 10-fold in LB medium in a 96-well plate. Cells were spotted using a 96-well replica plater onto the indicated media and incubated at 37 °C overnight.

### 1-N-phenylnapthylamine (NPN) uptake assay
The NPN assay was performed as described previously[17]. Overnight cultures were diluted and grown to $OD_{600} = 0.6$-$0.8$, and then normalized to $OD_{600} = 0.5$. Cells were washed and resuspended in 5 mM HEPES buffer (pH 7.2). Two hundred microliters of cells were aliquoted into a clear-bottom black-well plate (Corning Incorporated, Cat. #3603). NPN was added at a final concentration of 10 μM. The plate was sealed with a Breathe-Easy gas-permeable membrane (Sigma-Aldrich, Cat. #Z380059) and incubated in a Synergy H1 microplate

reader (BioTek). Fluorescence (excitation 350 nm, emission 420 nm) was measured after 20 min of incubation.

## Immunoblot analyses

Exponentially growing cells were normalized to $OD_{600} = 1$ and harvested by centrifugation. Cell pellets were resuspended in lysis buffer (25 mM Tris [pH 6.8], 1% SDS) and boiled for 5 min, followed by centrifugation. Supernatant was collected and protein concentration was measured using the Pierce BCA protein assay kit (ThermoFisher, Cat. #23225). Ten micrograms of protein were diluted in 4X Laemelli sample buffer (Bio-Rad, Cat. #1610747) supplemented with β-mercaptoethanol (except LptD-oxidized samples, for which β-mercaptoethanol was left out), and boiled for 10 min. Samples were loaded on an 8% (LptD samples) or 10% (all others) SDS-polyacrylamide gel, and electrophoresed. Proteins were transferred onto a nitro-cellulose membrane using the Trans-Blot Turbo Transfer system (BioRad, Cat. #1704150) and membranes were blocked for 1 h in 5% milk at room temperature. Immunoblotting was performed using polyclonal antisera against BamA (1:25,000 dilution), OmpF/C/A (1:25,000 dilution), LamB/MBP (1:12,500 dilution), or monoclonal antibody recognizing GroEL (1:50,000, Sigma-Aldrich, Cat. #G6532) or FLAG (1:5000 dilution, Sigma-Aldrich, Cat. #F3165), followed by secondary antibodies goat anti-rabbit IgG horseradish peroxidase (1:10,000 dilution, Sigma-Aldrich, Cat. #12-348) or goat anti-mouse IgG horseradish peroxidase (1:10,000 dilution, BioRad, Cat. #1706516).

## LPS analysis

LPS levels were analyzed as previously described[8]. Cells were grown to exponential phase and an $OD_{600} = 1$ culture was obtained by centrifugation. The cell pellet was washed with PBS and resuspended in 100 μL of 1X NuPAGE LDS sample buffer (ThermoFisher, Cat. #NP0007) supplemented with 4% β-mercaptoethanol. Samples were boiled for 10 min and 2 μL of 20 mg/ml proteinase K (New England Biolabs, Cat. #P8107S) were added. Samples were incubated at 55 °C for 16 h, followed by inactivation by boiling for 5 min. Lysates were loaded onto a 4–12% NuPAGE Bis-Tris gradient gel (ThermoFisher, Cat. #NP0322BOX) and resolved by SDS-PAGE. Gels were stained with Pro-Q Emerald 300 Lipopolysaccharide Gel Stain kit (ThermoFisher, Cat. #P20495) using manufacturer's instructions. LPS bands were visualized by UV transillumination.

## Fractionation of OM vesicles (OMVs)

OMVs were collected as described previously[8]. Briefly, overnight cultures were diluted 1:100 into fresh LB and grown to $OD_{600} = 0.6$-0.8. To collect whole-cell lysate controls, an $OD_{600} = 1$ culture was obtained by centrifugation, resuspended in 50 μL of 2X Laemmli buffer (BioRad, Cat. #1610747) with 10% β-mercaptoethanol (MilliporeSigma, Cat. #M3148), and boiled for 10 min. To collect OMVs, an $OD_{600} = 3$ culture was obtained by centrifugation. Supernatants were collected and adjusted to equal volume with fresh LB. Supernatants were filtered through a 0.2-μm filter (MilliporeSigma, Cat. #SLGPR33RB) and then through an Amicon Ultra-15 centrifugal filter with 100 kDa molecular weight cutoff (MilliporeSigma, Cat. #UFC9100) to isolate and concentrate OMVs. Centrifuged samples were adjusted to equal volume and resuspended in 2X Laemmli buffer (BioRad, Cat. #1610747) supplemented with 10% β-mercaptoethanol (MilliporeSigma, Cat. #M3148). Samples were boiled for 10 min, loaded on a 10% SDS-polyacrylamide gel, and processed as described above by immunoblot analysis.

## Single-cell imaging

Overnight cultures were diluted 1:200 (or 1:100 for ΔbamD cultures due to their low cell density) and grown at 37 °C, followed by two 1:10 dilutions (at $OD_{600}$ around 0.1) to generate steady-state log-phase

cells. Cells were imaged on a Nikon Eclipse Ti-E inverted fluorescence microscope with a 100X (NA 1.40) oil-immersion objective (Nikon Instruments). Images were collected on a Zyla 5.5 sCMOS camera (Andor Technology) using μManager v. 1.4[37]. Cells were maintained at 37 °C during imaging with an active-control environmental chamber (HaisonTech).

For single-cell imaging on agarose pads, 1 μL of cells was spotted onto a pad of 1% agarose in fresh LB. For microfluidic flow-cell experiments, including the spent medium assays, the oscillatory osmotic shock assays, and the plasmolysis/lysis assays, cells were loaded into CellASIC ONIX microfluidic flow cells (Sigma-Aldrich, Cat. #B04A-03-5PK) and medium was exchanged using the CellASIC ONIX2 microfluidic platform (Sigma-Aldrich, Cat. #CAX2-S0000).

## Imaging in microfluidic flow cells

ONIX B04A plates were loaded with medium and pre-warmed to 37 °C. Exponentially growing cells were incubated at 37 °C in fresh LB without shaking for 60 min before transitioning into experimental conditions and imaging. For spent medium assays, cells were perfused with spent LB subsequent to fresh LB. For oscillatory osmotic shock assays, cells were subjected to alternating hyperosmotic shocks of 400 mM sorbitol (Sigma-Aldrich, Cat. #S1876, dissolved in LB) and recovery without sorbitol, with a period of 2 min (1 min shock and 1 min recovery). For plasmolysis/lysis assays to measure OM stiffness, cells were subjected first to hyperosmotic shock with 1 M sorbitol in LB and then treated with 5% N-lauroylsarcosine sodium salt (MP biomedicals, Cat. #190289) to remove the OM. Cells were stained with 300 μM 3-[[(7-Hydroxy-2-oxo-2H-1-benzopyran-3-yl)carbonyl]amino]-D-alanine hydrocholoride (HADA, MedChemExpress, Cat. #HY-131045/CS-0124027) and perfused with 0.85X PBS prior to the osmotic shock.

## Spheroplast induction

Cells from a log-phase culture were loaded into an ONIX B04A plate and perfused with fresh LB and 2 μg/mL FM 4-64 (Invitrogen, Cat. #T13320) for 1 h, followed by 1 M sorbitol in LB for 10 min to induce plasmolysis. Plasmolyzed cells were treated with 1 mg/mL lysozyme (Sigma-Aldrich, Cat. #L6876) and 1 M sorbitol in LB to digest the cell wall, creating spheroplasts within 15 min of lysozyme treatment. Spheroplast cell contours were measured immediately after the loss of rod shape to minimize the impact of subsequent growth.

## Image analysis

Time-lapse images were first segmented with the software *DeepCell*[38], and the resulting segmented images were analyzed using *Morphometrics*[39] to obtain single-cell lineages and cell contours at sub-pixel resolution. Cell width and length were calculated using the MicrobeTracker meshing algorithm[40]. For plasmolysis/lysis assays, image segmentation and cell length measurements were performed using *Morphometrics*.

## Single-cell growth rate and morphology analysis

The instantaneous growth rate of individual cells was computed using cell contour and cell lineage data obtained using *Morphometrics*. The two-dimensional projected cell area $A$ was used as a proxy for cell biomass. At each time point, instantaneous growth rate was calculated from a local linear fit of $\ln(A)$. We determined the longitudinal axis of each cell contour and measured local cell width at equally spaced locations along the longitudinal axis. The coefficient of variation (CV) of cell width was defined for each cell as the ratio of the standard deviation to the mean of local width measurements along the rod-shaped cell body (measurements at the poles were excluded based on the curved geometry).

## Calculation of OM rest length and estimation of OM stiffness

To calculate OM rest length, we used *Morphometrics* to measure HADA-labeled cell-envelope contours before and after plasmolysis and FM 4-64-labled OM contours of wall-less spheroplasts induced by lysozyme as described above. For each cell, we calculated the surface area of the turgid cell, $A_{turgid}$, from its contour by assuming rotational symmetry around its long axis. After digestion of the cell wall by lysozyme, the OM adopted an amorphous shape, while remaining trapped in the flow cell with height assumed to be equal to the cell width before plasmolysis, $w$. We calculated the surface area of the OM, $A_{OM}$, using its two-dimensional area, $a$, and contour length, $s$, via the relation $A_{OM} = 2a + ws$. Finally, to calculate the rest length of the OM, $l_{OM}$, defined as the length of the OM if it were reshaped into a rod cell shape of width $w$, we used the equation $l_{OM} = l_{turgid} + (A_{OM} - A_{turgid})/(\pi w)$. Here, we approximated a rod-shaped cell as a cylinder capped by two hemispheres. Thus, the difference between the length of a turgid cell and the OM rest length is proportional to the difference between the corresponding surface areas by a factor of $1/(\pi w)$, the reciprocal of the cross-sectional perimeter.

OM stiffness was estimated using a plasmolysis-lysis assay[3] performed in a CellASIC ONIX microfluidic flow cell (Sigma-Aldrich, Cat. #B04A-03-5PK). Cell length was measured in the turgid state ($l_1$), under plasmolysis ($l_2$), and after lysis ($l_3$). Length contractions between these states were calculated as $\varepsilon_{12} = (l_1 - l_2)/l_2$, $\varepsilon_{23} = (l_2 - l_3)/l_3$, and $\varepsilon_{13} = (l_1 - l_3)/l_3$. The OM stiffness $k_{OM}$ relative to that of the cell wall $k_{CW}$ was estimated using the average value of the length contractions as $k_{OM} = k_{CW} \langle\varepsilon_{23}\rangle / [\langle\varepsilon_{12}\rangle (1 + \langle\varepsilon_{23}\rangle)]$, as in ref. 3.

## Reporting summary

Further information on research design is available in the Nature Portfolio Reporting Summary linked to this article.

## Data availability

All data used for generating figures in this study are available at the Stanford Digital Repository: https://purl.stanford.edu/gp172gf6221. Source data are provided with this paper.

## Code availability

All code used for generating figures in this study are available at the Stanford Digital Repository: https://purl.stanford.edu/gp172gf6221.

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

## Acknowledgements

We thank members of the Huang and Silhavy labs for helpful discussions. This work was funded by National Science Foundation grants EF-2125383 (to K.C.H.) and IOS-2032985 (to K.C.H.) and the National Institute of General Medical Sciences of the National Institutes of Health under Awards 5R35GM118024 (to T.J.S.) and F32GM139232 (to I.V.M.). The content is solely the responsibility of the authors and does not necessarily represent the official views of the National Institutes of Health. K.C.H. is a Chan Zuckerberg Biohub Investigator.

## Author contributions

I.V.M., J.S., K.C.H., and T.J.S. designed the research; I.V.M. and J.S. performed the research; I.V.M., J.S., K.C.H., and T.J.S. analyzed data; and I.V.M., J.S., K.C.H., and T.J.S. wrote the paper.

## Competing interests

The authors declare no competing interests.
