## [Peer Review File · Nature Communications]

Mechanism of outer membrane destabilization by global reduction of protein contentReviewer #1 (Remarks to the Author):

The manuscript by Silhavy and co-workers takes advantage of a Bam mutant in which a gain-of-function mutation in *bamA* allows deletion of the essential *bamD* lipoprotein. The resulting Δ *bamD* mutant displays lysis phenotype in stationary phase as a consequence of severe destabilization of the OM and envelope rupture. Δ *bamD* mutant cells show a severely reduced OMPs level and shape defects, compared to the parental strain. While the total LPS content is comparable between Δ *bamD* mutant and parental strain, the LPS profile changes between the two strains. Lysis phenotype, growth defects but not shape irregularities of Δ *bamD* mutant are suppressed by deletion of *mlaA* and *pldA* implicated in removing or destroying, phospholipids from the OM outer leaflet, respectively. These observations suggest that in Δ *bamD* mutant prevention of lysis depends on the level of phospholipids at the OM outer leaflet. However, increased level of phospholipids at the OM outer leaflet does not rescue cell viability under osmotic shocks suggesting that OM stiffness in Δ *bamD* Δ *mlaA* Δ *pldA* mutant is still compromised. Overall, the authors conclude that is the decreased OMPs level and the altered OMP network in the OM that mechanically destabilizes the OM pointing to an important role of the OM not only to the overall mechanical properties of the cell envelope but also in maintaining the cell shape. The manuscript nicely combines a very elegant genetic approach with advanced biophysical techniques to dissect the mechanical properties of the cell envelope and to understand at the molecular level the contribution of OM components/component network to the OM stiffness. My few comments are listed below:

Lines 120-124 and Fig. 2B

I understand that instantaneous growth rate indicates the increase in cell size in the time unit, what is not clear is how the coefficient of variation CV is calculated (it is not described in the figure legend nor in the Methods). From Fig. 2B it seems that instantaneous growth rate is significantly lower in the Δ *bamD* cells compared to the parental *bamAE470K* cells irrespective of the CV parameter. Significance is shown between pairs of Δ *bamD* and *bamAE470K* with a given morphology (CV) but not across pairs with different morphology (CV) as indicated in the text. Can the authors explain/comment?

Lane 127

Based on data reported in Fig. 2 the authors conclude that disrupting the BAM complex has a global effect on cell morphology. Have these morphology defects been observed in other Bam mutants as well?

Line 159 and Figure 3AB

All *bamD* deleted mutants, irrespective of the deletion of *pdlA* or *mlaA*, display an altered LPS profile. Can the authors comment? Do the authors know what is the chemical nature of these modifications? Is it possible that the alteration of the LPS profile observed in all *bamD* mutants also contributes to the lysis phenotype in stationary phase? In other words, can the author exclude that the observed LPS modifications may alter the mechanical properties of the OM?

Line 194 Fig. S1C

Fig S1C shows lysis in spent medium of Δ *bamD* *lpxC101* Δ *pdlA* cells and not of Δ *bamD* *lpxC101* Δ *mlaA* as indicated in the text.

Line 200

The authors claim that in the mutants (Δ *mlaA*, Δ *pdlA* and Δ *mlaA*- Δ *pdlA*) that suppress the lysis phenotype the level of OMPs does not change compared to the parental Δ *bamD* mutant. However, as shown in Fig. 3A, the Δ *mlaA*, Δ *pdlA* and Δ *mlaA*- Δ *pdlA* suppressor mutants appear to have an increased level of BamA and OmpC/F. Only in the *lpxC101* suppressor mutant the level of OMPs is comparable to that of the parental Δ *bamD* mutant. Can the authors comment on that?

Lanes 302-307

Based on results shown in Fig. 2 and 4 the authors conclude that in wild type strains a stiff OM contributes to the maintenance of cell shape and prevents the development of irregular shapes. Can the authors exclude PG remodeling in the Δ *bamD* mutant and its derivatives in response to the OM alterations as a factor contributing to irregular cell shapes?

Minor comments

Figure 2: there are 2 panels named "D", one should be "E".

Table S1 appears to be missing in the supplemental material.

Line 194

"spend media" should be "spent medium"

Line 202

"modified" should be "decreased"

Lines 229, and 232

Fig. 5B should be Fig. 4B

Line 293

Fig. 5 should be Fig. 4

There is no reference in the text to Fig. 5 (Model of OM destabilization in bamD deleted cells)

Reviewer #2 (Remarks to the Author):

In this article, the authors take advantage of the fact that BamD is non essential in bamAE470 cells to investigate the impact of bamD deletion on cellular physiology. They found that deleting bamD impacts cell growth and causes cells to lyse during stationary phase or when resuspended in spent medium. bamD deletion also decreases OM integrity and alters cell shape. Using microfluidics and monitoring the release of fluorescent markers, the authors also show that the OM ruptures before the IM. As expected, bamD deletion was also found to decrease the levels of OMPs, altering OM composition. It is generally considered that when OMPs levels are down, phospholipids accumulate in the outer leaflet of the OM to fill the voids. Accordingly, the authors found that deleting pldA, a phospholipase that degrades phospholipids, and mlaA, a component of the mla system that transports phospholipids from the OM to the IM, abolished lysis. Deleting pldA and introducing the lpxC101 allele (which decreases LPS synthesis) also prevented lysis. Interestingly, the authors found that rescue was not due to restored OMP levels but rather to increased phospholipids levels in the OM. Phospholipids accumulation in the OM does not however restore cell shape or OM mechanical integrity.

Overall, the paper is nicely written and interesting. Here are my comments:

1-the authors should discuss why cells specifically lyse in stationary phase.

2-AsmA proteins have been shown to transport PLs to the OM. What is the impact of asmA deletion in the Δ bamD mutant?

3-why do the authors think that filling the voids when OMPs are down necessarily involves flipping phospholipids from the inner leaflet to the outer leaflet: one could fill the voids with LPS in the outer leaflet and PL in the inner leaflet. Could the authors comment on that? (I agree that the results obtained by deleting mlaA and pldA provide indirect evidence for the accumulation of PLs in the outer leaflet)

4- To support their main claim, could the authors compare the proportion of PLs in the outer leaflet of the OM with or without bamD? One option could be to perform AFM-scanning to detect PLs islands, like in Benn et al.

5. What are we looking at in Fig.3A and B? Log or stat phase protein content? see lanes 154-155.

6-Line 285: how do they know flipping happens slowly

7-How do the authors explain the accumulation of modified LPS in Fig. 3B. Do these forms accumulate in the IM?

8-It would be interesting to test the OM in *bamA101* cells, in which OMP levels are also down. The results should be like those from Δ *bamD* cells. Analysing Δ *bamB* cells, in which major OMPs are down, but not LptD and BamA, would also be interesting to further test the model.

9-what about the levels of LpoA and LpoB in Δ *bamD* cells? If they are down, this could also explain the cell shape defects.

10-Lanes 229 and 232: Fig 4B instead 5B

REVIEWER COMMENTS

Reviewer #1 (Remarks to the Author):

The manuscript by Silhavy and co-workers takes advantage of a Bam mutant in which a gain-of-function mutation in bamA allows deletion of the essential bamD lipoprotein. The resulting Δ bamD mutant displays lysis phenotype in stationary phase as a consequence of severe destabilization of the OM and envelope rupture. Δ bamD mutant cells show a severely reduced OMPs level and shape defects, compared to the parental strain. While the total LPS content is comparable between Δ bamD mutant and parental strain, the LPS profile changes between the two strains. Lysis phenotype, growth defects but not shape irregularities of Δ bamD mutant are suppressed by deletion of mlaA and pldA implicated in removing or destroying phospholipids from the OM outer leaflet, respectively. These observations suggest that in Δ bamD mutant prevention of lysis depends on the level of phospholipids at the OM outer leaflet. However, increased level of phospholipids at the OM outer leaflet does not rescue cell viability under osmotic shocks suggesting that OM stiffness in Δ bamD Δ mlaA Δ pldA mutant is still compromised. Overall, the authors conclude that is the decreased OMPs level and the altered OMP network in the OM that mechanically destabilizes the OM pointing to an important role of the OM not only to the overall mechanical properties of the cell envelope but also in maintaining the cell shape.

The manuscript nicely combines a very elegant genetic approach with advanced biophysical techniques to dissect the mechanical properties of the cell envelope and to understand at the molecular level the contribution of OM components/component network to the OM stiffness.

We appreciate the support and helpful comments of the reviewer!

My few comments are listed below:

Lines 120-124 and Fig. 2B

I understand that instantaneous growth rate indicates the increase in cell size in the time unit, what is not clear is how the coefficient of variation CV is calculated (it is not described in the figure legend nor in the Methods). From Fig. 2B it seems that instantaneous growth rate is significantly lower in the Δ bamD cells compared to the parental bamAE470K cells irrespective of the CV parameter. Significance is shown between pairs of Δ bamD and bamAE470K with a given morphology (CV) but not across pairs with different morphology (CV) as indicated in the text. Can the authors explain/comment?

As described in the Methods, we used the Matlab package *Morphometrics* to determine the longitudinal axis of each cell contour and measured the width at equally spaced

locations along the cell length. The coefficient of variation (CV) of cell width is calculated for each individual cell as the standard deviation of width measurements along the rod-shaped cell body divided by the mean width. CV was used as an indicator of shape irregularity. We have added more details about this analysis in the Methods section (see lines 486-495).

We agree that showing the significance of differences between morphology groups would be useful and have added those indications to Fig. 2B.

Lane 127

Based on data reported in Fig. 2 the authors conclude that disrupting the BAM complex has a global effect on cell morphology. Have these morphology defects been observed in other Bam mutants as well?

Thank you for the interesting question. We quantified the morphology of two other mutants that alter the BAM complex, *bamA101* and Δ *bamB*. Both mutants showed disrupted morphology compared with *bamA^{E470K}* during log-phase growth, as indicated by the distribution of CV of cell width (see below), suggesting that disruption of the BAM complex generally perturbs cell shape regulation. We have included this plot in as Fig. S2A in the results section.

Line 159 and Figure 3AB

All *bamD* deleted mutants, irrespective of the deletion of *pdlA* or *mlaA*, display an altered LPS profile. Can the authors comment? Do the authors know what is the chemical nature of these modifications? Is it possible that the alteration of the LPS profile observed in all *bamD* mutants also contributes to the lysis phenotype in stationary phase? In other words, can the author exclude that the observed LPS modifications may alter the mechanical properties of the OM?

Modified LPS (M-LPS) is primarily LPS with colanic acid attached by the O antigen ligase. The $\Delta bamD$ mutant synthesizes the colanic acid capsule (the colonies are mucoid) regulated by the Rcs pathway that senses disruption to the OM integrity. Since K12 strains do not make O antigen, the O antigen ligase attaches colanic acid to LPS, making M-LPS. We do not think that LPS modification is a major contributor to cell lysis, as M-LPS is also present in the suppressors (Fig. 3B). Indeed, as shown in the figure below, removing the Rcs pathway in the $\Delta bamD \Delta pdlA$ strain by deleting either *rscF* or *rscB* did not alter the cell lysis phenotype. Hence, we conclude that M-LPS is unlikely to be a major factor in determining the mechanical properties of the OM. We have added these data as Figure S4 and the above discussion to the text in lines 211-216.

Figure: Disruption of the Rcs pathway through deletion of *rscB* or *rscF* does not reduce lysis. Black: $\Delta bamD \Delta pdlA$, light blue: $\Delta bamD \Delta pdlA rcsB::kan$, purple: $\Delta bamD \Delta pdlA rcsF::kan$. A) Disruption of the Rcs pathway in $\Delta bamD$ does not reduce cell lysis in stationary culture. B) Disruption of the Rcs pathway in $\Delta bamD$ does not reduce cell lysis after resuspension in spent medium.

Line 194 Fig. S1C

Fig S1C shows lysis in spent medium of $\Delta bamD lpxC101 \Delta pdlA$ cells and not of $\Delta bamD lpxC101 \Delta mlaA$ as indicated in the text.

Thanks for pointing out this issue. There was a typo in the figure reference in lines 202-204 (lines 194-195 in the manuscript before revision), which was supposed to reference Fig. S1B. We have fixed this error by switching panels B and C. (Note that this Figure has become Fig. S3).

Line 200

*The authors claim that in the mutants ($\Delta mlaA$, $\Delta pldA$ and $\Delta mlaA-\Delta pldA$) that suppress the lysis phenotype the level of OMPs does not change compared to the parental $\Delta bamD$ mutant. However, as shown in Fig. 3A, the $\Delta mlaA$, $\Delta pldA$ and $\Delta mlaA-\Delta pldA$ suppressor mutants appear to have an increased level of BamA and OmpC/F. Only in the *lpxC101* suppressor mutant the level of OMPs is comparable to that of the parental $\Delta bamD$ mutant. Can the authors comment on that?*

Thank you for your attention to detail! While it is true that the levels of BamA and OmpC/F slightly increased in the suppressor mutants, they did not return to wild-type levels, which was the comparison we intending to draw in the text. Hence, we believe that the suppression of lysis in the double mutant is due to the stabilized phospholipids in the membrane, instead of the slight increase in protein content. We have modified the text about this point (see lines 207-209).

Lanes 302-307

Based on results shown in Fig. 2 and 4 the authors conclude that in wild type strains a stiff OM contributes to the maintenance of cell shape and prevents the development of irregular shapes. Can the authors exclude PG remodeling in the $\Delta bamD$ mutant and its derivatives in response to the OM alterations as a factor contributing to irregular cell shapes?

Although we do not have any evidence pointing to PG remodeling, we agree that it is a possible explanation for the altered cell shape in the $\Delta bamD$ mutant and its derivatives. If PG remodeling does occur and contributes to the irregular cell shapes, we propose that it is likely a downstream effect of a disrupted OM, which has very low stiffness in mutants carrying the *bamD* deletion.

Minor comments

Figure 2: there are 2 panels named "D", one should be "E".

Thank you, we have fixed this error.

Table S1 appears to be missing in the supplemental material.

We apologize, we have now included Table S1.

Line 194

“spend media” should be “spent medium”

Thank you, we have fixed the typo.

Line 202

“modified” should be “decreased”

We have made this change.

Lines 229, and 232

Fig. 5B should be Fig. 4B

Thank you, we have fixed this error.

Line 293

Fig. 5 should be Fig. 4

We meant for this reference to be a callout to the model in Fig. 5.

There is no reference in the text to Fig. 5 (Model of OM destabilization in bamD deleted cells)

We have added reference to Fig. 5 in the discussion.

Reviewer #2 (Remarks to the Author):

In this article, the authors take advantage of the fact that BamD is non essential in bamAE470 cells to investigate the impact of bamD deletion on cellular physiology. They found that deleting bamD impacts cell growth and causes cells to lyse during stationary phase or when resuspended in spent medium. bamD deletion also decreases OM integrity and alters cell shape. Using microfluidics and monitoring the release of fluorescent markers, the authors also show that the OM ruptures before the IM. As expected, bamD deletion was also found to decrease the levels of OMPs, altering OM composition. It is generally considered that when OMPs levels are down, phospholipids accumulate in the outer leaflet of the OM to fill the voids. Accordingly, the authors found that deleting pldA, a phospholipase that degrades phospholipids, and mlaA, a component of the mla system that transports phospholipids from the OM to the IM, abolished lysis. Deleting pldA and introducing the lpxC101 allele (which decreases LPS synthesis) also

prevented lysis. Interestingly, the authors found that rescue was not due to restored OMP levels but rather to increased phospholipids levels in the OM. Phospholipids accumulation in the OM does not however restore cell shape or OM mechanical integrity.

Overall, the paper is nicely written and interesting. Here are my comments:

We appreciate the reviewer's support and helpful comments!

1-the authors should discuss why cells specifically lyse in stationary phase.

The lysis in stationary phase cultures or spent medium is mostly due to the depletion of Mg^{2+} , which disrupts LPS-LPS interactions, leading to weakening of the OM. Indeed, we were able to prevent cell lysis by supplementing spent LB with Mg^{2+} , as shown in the figure below. We have added this figure as new Figure S1 and added additional discussion in lines 99-102.

Figure: Mg^{2+} addition prevents cell lysis in spent medium. Black: wild type (*bamA^{E470K}*), red: $\Delta bamD$, blue: $\Delta bamD + 5 \text{ mM } Mg^{2+}$.

2-AsmA proteins have been shown to transport PLs to the OM. What is the impact of *asmA* deletion in the $\Delta bamD$ mutant?

We attempted to examine the effects of deleting *yhdP* and *tamB* in the $\Delta bamD$ mutant, but the high redundancy of AsmA proteins as shown in recent studies made

interpretation of the results complicated. Moreover, the deletion of *asmA* has been shown to impact LPS levels, further complicating the interpretation. While we appreciate the suggestion from the reviewer, we believe that further studies are needed to better understand the impact of *asmA* deletion in the $\Delta bamD$ mutant.

*3-why do the authors think that filling the voids when OMPs are down necessarily involves flipping phospholipids from the inner leaflet to the outer leaflet: one could fill the voids with LPS in the outer leaflet and PL in the inner leaflet. Could the authors comment on that? (I agree that the results obtained by deleting *miaA* and *pldA* provide indirect evidence for the accumulation of PLs in the outer leaflet)*

Filling the voids with LPS in the outer leaflet is possible but requires an increase in LPS synthesis, which we did not observe (Fig. 3B). We think the results from the deletion of *miaA* and *pldA* provide strong evidence for the translocation of PLs to the OM outer leaflet, since the direct effect of *pldA* deletion is to allow PLs to stay in the outer leaflet, which is required for the suppression of cell lysis (Fig. 3C,D). This interpretation has been used to support the above point in other published studies, such as Powers and Trent (2018) (ref. 18 in main text).

*4- To support their main claim, could the authors compare the proportion of PLs in the outer leaflet of the OM with or without *bamD*? One option could be to perform AFM-scanning to detect PLs islands, like in Benn et al.*

We appreciate the reviewer's suggestion. However, we believe that AFM is not a suitable method for this purpose for two reasons. First, as demonstrated by Benn et al., when the OMP lattice network is disrupted, contrast is lost, meaning that non-network regions cannot be distinguished. Second, the OM of the $\Delta bamD$ mutant is so fragile that AFM scanning would likely disrupt the membrane or even break it.

5. What are we looking at in Fig.3A and B? Log or stat phase protein content? see lanes 154-155.

Fig. 3A,B focus on exponentially growing cells. Stationary phase protein content is difficult to interpret due to the contribution of lysing cells. We have now added this information to the figure legends as well.

6-Line 285: how do they know flipping happens slowly

The presence of PLs in the outer leaflet of the OM disrupts the permeability barrier, and the flipping of PLs from the inner to outer leaflet is not catalyzed by enzymes. Additionally, movement of the polar PL headgroup through the hydrophobic

membrane interior is not thermodynamically favorable, which suggests that the reaction occurs slowly. This interpretation is supported by other published papers that also assume slow movement of PLs across the membrane, such as Nagy et al. (2019) (reference 19 of the main text).

7-How do the authors explain the accumulation of modified LPS in Fig. 3B. Do these forms accumulate in the IM?

Modified LPS (M-LPS) is primarily LPS with colanic acid attached by the O antigen ligase. The $\Delta bamD$ mutant synthesizes the colanic acid capsule (the colonies are mucoid) regulated by the Rcs pathway that senses disruption to the OM integrity. Since K12 strains do not make O antigen, the O antigen ligase attaches colanic acid to LPS, making M-LPS. We do not think that LPS modification is a major contributor to cell lysis, as M-LPS is also present in the suppressors (Fig. 3B). Indeed, as shown in the figure below, removing the Rcs pathway in the $\Delta bamD \Delta pldA$ strain by deleting either *rscF* or *rscB* did not alter the cell lysis phenotype. Hence, we conclude that M-LPS is unlikely to be a major factor in determining the mechanical properties of the OM. Moreover, it is unlikely for M-LPS to accumulate in the IM, as LPS accumulation in the IM is highly toxic. We have added these data as Figure S4 and the above discussion to the text in lines 211-216.

Figure: Disruption of the Rcs pathway through deletion of *rscB* or *rscF* does not reduce lysis. Black: $\Delta bamD \Delta pldA$, light blue: $\Delta bamD \Delta pldA rcsB::kan$, purple: $\Delta bamD \Delta pldA rcsF::kan$. A) Disruption of the Rcs pathway in $\Delta bamD$ does not reduce cell lysis in stationary culture. B) Disruption of the Rcs pathway in $\Delta bamD$ does not reduce cell lysis after resuspension in spent medium.

8-It would be interesting to test the OM in *bamA101* cells, in which OMP levels are also down. The results should be like those from Δ *bamD* cells. Analysing Δ *bamB* cells, in which major OMPs are down, but not *LptD* and *BamA*, would also be interesting to further test the model.

Thanks for the interesting suggestion. We quantified the OM stiffness and log phase morphology of the *bamA101* and Δ *bamB* mutants (see below) The plasmolysis/lysis assay showed that both mutants have a comparably stiff OM as *bamA^{E470K}*, as indicated by similar distributions of ϵ_{12} (length contraction from a turgid cell to a plasmolyzed cell) and ϵ_{23} (length contraction from a plasmolyzed cell to a lysed cell). This observation likely reflects that the extent of OMP reduction in both mutants are much less severe than in the Δ *bamD* mutant. Interestingly, both mutants showed disrupted morphology compared with *bamA^{E470K}* during log-phase growth in a microfluidic flow cell, as indicated by the large tail in which CV>10%, suggesting that maintaining a stiff OM is not sufficient for cell width regulation. We have added these data as Figure S2.

9-what about the levels of *LpoA* and *LpoB* in Δ *bamD* cells? If they are down, this could also explain the cell shape defects.

We have not specifically examined *LpoA* and *LpoB* levels, as we do not have any reason to expect them to be down. We agree that examining the levels of a variety of proteins that are associated with cell shape such as *LpoA* and *LpoB* could provide insight into lipoprotein content in the Δ *bamD* mutant and their connection to shape defects. Such an investigation would require substantial follow-up experiments that would be far beyond the scope of this study (which does not focus on the cell shape defects). We plan to follow up on this interesting question in future studies.

10-Lanes 229 and 232: Fig 4B instead 5B

Thank you, we have fixed this error.